# In-Situ Text-Only Adaptation of Speech Models with Low-Overhead Speech Imputations

**Ashish Mittal**
IBM Research, IIT Bombay
arakeshk@in.ibm.com

**Sunita Sarawagi & Preethi Jyothi**
IIT Bombay
{sunita,pjyothi}@cse.iitb.ac.in

## Abstract

Fast and accurate adaptation of automatic speech recognition (ASR) systems using only text data in the target domain is a problem of long-standing practical relevance. Text-only adaptation was easy in traditional cascaded ASR systems with completely decoupled acoustic and language models. Recently, the RNN-Transducer (RNN-T) has emerged as a popular ASR model because of its high accuracy, low latency, and capability of supporting streaming input. However text-only adaptation of the RNN-T model is significantly more challenging due to its tight integration of acoustic and language models and end-to-end training. Existing recent approaches for text-only adaptation of RNN-Ts, either entail significant modification to the network or introduce high latency during decoding. We propose a new approach (TOLSTOI) that using text imputes speech representations internal to the ASR model, and performs in-situ adaptation that results in higher adaptation accuracy without any runtime overheads during decoding. Our imputation model is a function of the labeled data and trained parameters of the ASR model, and that we show, is more effective in controlling catastrophic forgetting compared to existing methods. We establish the effectiveness of TOLSTOI using three target domains and two ASR models of varying complexity. We yield up to 35% relative reduction in word error rate with text only adaptation, while forgetting the least compared to existing adaptation approaches. Our method is easy to implement and can be harnessed on existing RNN-T models without requiring ASR model training from scratch.

## 1 Introduction

Text-only adaptation of end-to-end (E2E) automatic speech recognition (ASR) systems to new target domains is of much practical interest since in many situations, e.g. mobile phones, it is easier to get target-specific text data than the corresponding audio. Efficient and effective text-only adaptation remains an open problem in large part due to the nature of E2E ASR systems that use a single model to jointly learn both a mapping from speech to text and a language model (LM), thus rendering traditional LM adaptation techniques for ASR (Bellegarda, 2004) ineffective.

RNN-Transducer (RNN-T) models (Graves, 2012) are one of the most popular E2E ASR architectures that achieve high accuracy and enable real-time decoding of speech, thus making them the predominant choice for ASR on mobile devices (He et al., 2019). Customizing RNN-T models using text-only data has gathered momentum in recent years. For any ASR applications using RNN-T models, running at real-time is a critical requirement. Thus, we seek simple and accurate text-only adaptation techniques that do not increase the model complexity. In this work, we propose such an approach TOLSTOI that is simple in its design and works with pretrained RNN-T models while enabling fast and accurate adaptation to the target domain. We will first review existing approaches to the problem of text-only adaptation to help contextualize TOLSTOI better.

A popular solution for text-only adaptation of E2E ASR systems is shallow fusion (Hannun et al., 2014) where scores from the E2E model are combined with scores from an external LM trained on the target text during beam search decoding. While simple in its design, this technique significantly increases decoding time due to the reliance on an external LM during inference. More recent work on adapting RNN-T models using only text aims at directly updating the parameters of the prediction network (Pylkkonen et al., 2021; Chen et al., 2022a). However such techniques do

not yield very accurate adaptation to the target text and also involve architectural changes to the RNN-T that necessitate training the model from scratch. Text-only adaptation can also be tackled by generating corresponding speech via text-to-speech synthesis (TTS). The main limitations of TTS-based adaptation are significant computational costs and the reliance on high-quality TTS systems that are available only for a small subset of high-resource languages and accents.

From these prior works, the key requirements for practical text-only adaptation of ASR that emerge are i) the model should adapt to target-domain text-only data with high accuracy ii) the adaptation should be applied to existing pretrained models without any retraining iii) the inference should be fast and inexpensive and iv) the adaptation should not lead to catastrophic forgetting (Goodfellow et al., 2013; Takashima et al., 2022). We propose TOLSTOI that addresses all four requirements. Starting from text in the target domain, we impute speech representations as would have been produced by the transcription network of a pretrained RNN-T model. Our imputation model is a simple feedforward network (with roughly 200K parameters) that incurs minimal overhead in its training by harnessing forced alignments and representations from the ASR model. Using the trained imputation model, we generate sequences of speech representations for all the text in the target domain which are used for in-situ adaptation of the RNN-T ASR model. TOLSTOI can be used with any existing pretrained RNN-T. We do not introduce any new parameters in the RNN-T and do not rely on any external LMs, thus incurring no additional overhead on latency at inference time. Along with yielding fast and accurate adaptation to the target domain, TOLSTOI also safeguards against forgetting since the imputation model is trained to mimic representations from the source distribution. TOLSTOI yields up to 35% relative word error rate (WER) reduction on a new target domain, while maintaining the same decoding latency as the base RNN-T model and ensuring minimal forgetting of its source information when compared to three other competitive baselines. We also present a detailed ablation study to justify the various design choices of TOLSTOI.

## 2 RELATED WORK

**LM adaptation in traditional ASR systems** Unlike end-to-end models, traditional ASR systems adopt a cascaded structure with the LM being completely decoupled from the acoustic model (Mohri et al., 2002). This enables easier adaptation of the LM to a target domain (Hori et al., 2003; Bellegarda, 2004; Neubig et al., 2009; Gangireddy et al., 2016) and also allows for ASR lattice rescoring with an external LM (Park et al., 2010; Xu et al., 2018).

**LM fusion** A popular approach for text-only adaptation of end-to-end ASR is "shallow fusion" where an external LM is log-linearly interpolated with the RNN-T output during beam decoding Kannan et al. (2018). For RNN-T models, another recent approach is to extract internal LM probabilities and discount with the ratio of external and internal LM probabilities (McDermott et al., 2019; Meng et al., 2021a;b; Udagawa et al., 2022). These techniques incur a significant overhead at inference time due to the external LM and also require careful tuning of the interpolation weight used for the external LM.

**Synthesizing audio** Another approach to text-only adaptation is to synthesize audio using text-to-speech (TTS) synthesis (Zheng et al., 2021; Deng et al., 2021; Joshi & Singh, 2022; Hayashi et al., 2018; Hori et al., 2019; Baskar et al., 2021; Chen et al., 2022c). However, this is a slow generation process and relies on access to high-quality TTS Shen et al. (2018) which is absent for most languages. To address these issues, recent work on text-only adaptation has investigated generating simpler pseudo-speech representations called "textograms" by repeating one-hot encodings of the output labels for a fixed duration (Thomas et al., 2022). The input to the RNN-T is augmented to accept a textogram as an additional channel. This model requires training the RNN-T from scratch and also negatively impacts the decoding latency.

**Fine-tuning RNN-T model parameters** Recent approaches exploit the inherent structure of the RNN-T to perform in-situ text-only adaptation. Pylkkonen et al. (2021) adds a separate LM output head to the prediction network in an RNN-T (that handles text-only inputs) and both are jointly finetuned using the target text. Chen et al. (2022a) first factorize the prediction network into two networks that separately handle "blank" tokens (capturing alignment with the audio) and the output vocabulary tokens, before adapting the latter with the target text. This technique requires retraining the RNN-T model and does not yield accurate adaptation.

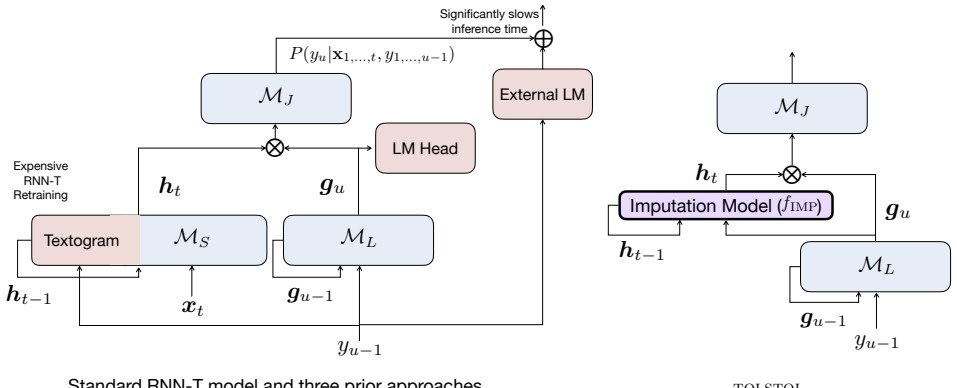

Figure 1: Figure on the left shows a schematic diagram of the standard RNN-T architecture (in blue). Three existing approaches for text-only adaptation are shown as appendages to the RNN-T model (in red): (1) "Textogram" increases the dimensionality of the input to the transcription network $\mathcal{M}_S$ thus requiring an expensive model retraining. (2) Shallow fusion uses an external LM during inference that leads to significant degradation of test-time latency. (3) A separate LM head can be added and jointly trained with the prediction network on the target text. This, however, does not result in very accurate adaptation. The figure on the right shows TOLSTOI that uses a lightweight imputation model to generate speech representations corresponding to the target text.

Figure 1 shows a technique from each of the above-mentioned categories and how it is integrated within the standard RNN-T architecture. (A recent line of work focuses on learning shared speech and text representations, thus allowing for the use of unpaired text (Bapna et al., 2021; Ao et al., 2021; Tang et al., 2022; Chen et al., 2022b). An extension of these ideas to streaming RNN-T models (Sainath et al., 2023) was concurrent to our work and would be interesting to explore further.)

## 3  OUR APPROACH (TOLSTOI)

We are given an ASR model $\mathcal{M}$ trained on a labeled dataset $D : \{(\boldsymbol{x}^1, \boldsymbol{y}^1) \ldots, (\boldsymbol{x}^N, \boldsymbol{y}^N)\}$ where $\boldsymbol{x}^i \in \mathcal{X}$, the space of speech utterances, and $\boldsymbol{y}^i \in \mathcal{Y}$, the space of text transcripts. Each speech utterance $\boldsymbol{x}^i$ comprises of a variable number of frames $\boldsymbol{x}_1^i, \ldots, \boldsymbol{x}_{T_i}^i$ where each $\boldsymbol{x}_t^i$ is a fixed-length real-valued vector denoting features such as spectrogram of the audio frame. Each text sequence comprises of a variable number of tokens $\boldsymbol{y}^i = (y_1^i, \ldots, y_{U_i}^i)$ where each $y_u \in \mathcal{V}$, the output vocabulary. Popular choices for the output vocabulary are characters and subwords (Sennrich et al., 2015). Typically the number of text tokens $U_i \ll T_i$, the number of audio frames. Let $\mathcal{P}(\mathcal{X}, \mathcal{Y})$ denote the distribution of speech and text from which the training data is sampled.

Our goal is to deploy the ASR model $\mathcal{M}$ on a target domain whose distribution $\tilde{\mathcal{P}}(\mathcal{X}, \mathcal{Y})$ differs from the training distribution. For the target domain, we only have text data $\tilde{D} = \{\tilde{\boldsymbol{y}}^1, \ldots, \tilde{\boldsymbol{y}}^k\}$ where the number of text samples in $\tilde{D}$ in the target is generally much smaller than the size of the training set $D$. Since we are only given text samples in the target distribution we assume that the training and target distributions differ only on the text marginals $\mathcal{P}(\mathcal{Y})$ and the distribution of the speech given the text stays unchanged. That is, $\mathcal{P}(\mathcal{X}|\mathcal{Y}) = \tilde{\mathcal{P}}(\mathcal{X}|\mathcal{Y})$. We seek to use $\tilde{D}$ to fine-tune the parameters of $\mathcal{M}$ so that the updated model $\tilde{\mathcal{M}}$ is accurate on speech corresponding to new text from the target distribution $\tilde{\mathcal{P}}$, without catastrophically deteriorating accuracy on samples from the training distribution $\mathcal{P}$. We propose to perform in-situ adaptation without introducing new layers or external LMs during deployment.

As mentioned earlier, we focus on adapting the RNN-Transducer (RNN-T) architecture (Graves, 2012; Graves et al., 2013) as the ASR model since it has recently emerged as a popular choice, particularly in mobile devices, because of its high accuracy, low latency, and capability of supporting streaming input. We first present a brief background of RNN-Ts.

**Background of RNN-T**  The RNN-T network comprises of three modules: (1) A speech module $\mathcal{M}_S$ with parameters $\boldsymbol{\theta}_S$ that converts speech frames $\boldsymbol{x} = \boldsymbol{x}_1 \ldots, \boldsymbol{x}_T$ to vectors $\boldsymbol{h}_1, \ldots, \boldsymbol{h}_T$ typ-

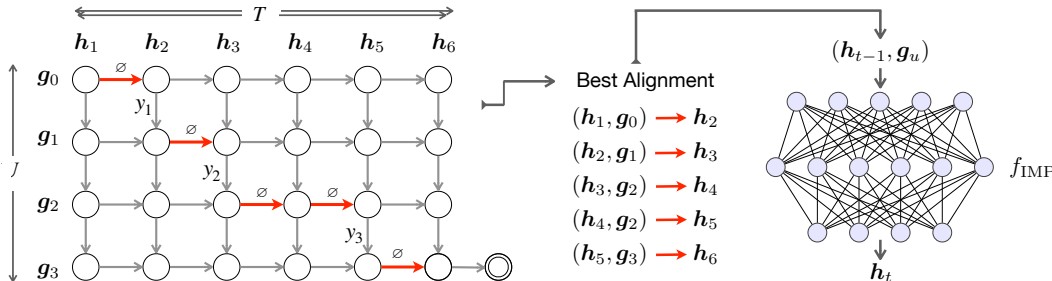

Figure 2: Training the imputation model of TOLSTOI using RNN-T alignments.

ically using a Transformer or recurrent network, (2) A language module $\mathcal{M}_L$ with parameters $\boldsymbol{\theta}_L$ that converts text tokens $\boldsymbol{y} = y_1, \ldots, y_U$ to contextual vectors $\boldsymbol{g}_1, \ldots, \boldsymbol{g}_U$ using a recurrent network so that $\boldsymbol{g}_u$ summarizes the tokens before $y_u$, and (3) A thin joint network $\mathcal{M}_J$ with parameters $\boldsymbol{\theta}_J$ that combines vectors $\boldsymbol{h}_t$ and $\boldsymbol{g}_u$ and outputs a softmax distribution spanning the vocabulary $\mathcal{V}$ plus the blank symbol $\varnothing$. To generate a token sequence $\boldsymbol{y}$, the RNN-T implicitly aligns each token $y_u$ to one frame $t$ where it is output. An example of a valid alignment[1] appears in Figure 2.

Even though the network is modular, all parameters are trained jointly end-to-end by maximizing the likelihood of the output $\boldsymbol{y}^i$ given speech $\boldsymbol{x}^i$ over all $(\boldsymbol{x}^i, \boldsymbol{y}^i) \in D$. During training, the log-likelihood of the target sequence $\boldsymbol{y}^i$ is computed by marginalizing over all possible alignments of the $T_i$ frames of $\boldsymbol{x}^i$ and the $U$ tokens of $\boldsymbol{y}$ using an efficient forward-backward algorithm; we will refer to this objective as the RNNT-loss (Graves, 2012).

$$\min_{\boldsymbol{\theta}_L, \boldsymbol{\theta}_S, \boldsymbol{\theta}_J} \sum_{(\boldsymbol{x}^i, \boldsymbol{y}^i) \in D} \text{RNNT-loss}(\boldsymbol{y}^i, \{\mathcal{M}_J(\cdot | \boldsymbol{g}_u = \mathcal{M}_L(y_1, \ldots, y_{u-1}), \boldsymbol{h}_t = \mathcal{M}_S(\boldsymbol{x}_1, \ldots, \boldsymbol{x}_T, t)\})$$

During inference, beam-search finds the best possible alignment $(t, u)$ and the predicted sequence is a concatenation of the non-blank tokens at each aligned $(t, u)$ (Graves, 2012; Saon et al., 2020).

**Motivation of our approach**  Figure 1 presents a schematic diagram of our imputation based adaptation. Given only text data $\tilde{D}$ for adaptation, we propose to update the language parameters $\boldsymbol{\theta}_L$ and joint parameters $\boldsymbol{\theta}_J$ while keeping the speech parameters $\theta_S$ fixed. This is challenging since even though the network architecture is modular, the training is not. Treating the $\mathcal{M}_L$ as a language model and updating part of the network using text-only data, as proposed in (Pylkkonen et al., 2021; Chen et al., 2022a), has the potential of deteriorating performance as the output vector of $\mathcal{M}_L$ gets incompatible with the vector from $\mathcal{M}_S$. We, therefore, propose to first augment the missing speech data in $\tilde{D}$ by imputing using a separate generator. However, training a full-fledged TTS model requires substantial training data and resources, and high-quality TTS models are not available for low-resource languages. Our key insight is to impute, not the full raw speech $\boldsymbol{x}$, but the $\boldsymbol{h}$ vectors from the last layer of the speech module $\mathcal{M}_S$. Generating the last layer vectors $\boldsymbol{h}$ is significantly easier than generating the raw audio signals $\boldsymbol{x}$. In fact, only a thin joint layer separates $\boldsymbol{h}$ from the output character distribution. So the $\boldsymbol{h}$ vectors are expected to be "closer" to text than speech. We are able to design a very simple and low-overhead model for imputing the $\boldsymbol{h}$ vectors from the text. In Section 3.1 we describe the design and training of the imputation model. Once the imputation model is trained, for any new target domain, we attach each text $\boldsymbol{y} \in \tilde{D}$ with imputed $\boldsymbol{h}$ values to create a proxy parallel dataset for fine-tuning $\mathcal{M}_J$ and $\mathcal{M}_L$. In Section 3.2 we describe how we perform this fine-tuning.

### 3.1 IMPUTATION MODEL

Let $\mathcal{H}$ denote the space of vectors from the last layer of $\mathcal{M}_S$. The goal of our imputation model is to directly model $\mathcal{P}(\mathcal{H}|\mathcal{Y})$ instead of $\mathcal{P}(\mathcal{X}|\mathcal{Y})$ that TTS models attempt to do. The imputation model generates proxy output vectors $\boldsymbol{h}_1, \ldots, \boldsymbol{h}_T$ of the speech module $\mathcal{M}_S$ given *only* a text sequence $y_1, \ldots y_U$ so as to mimic the output of the speech module $\mathcal{M}_S(\boldsymbol{x}_1, \ldots, \boldsymbol{x}_T)$ without having access to the real audio frames $\boldsymbol{x}_1, \ldots, \boldsymbol{x}_T$. A default option is to train a full-fledged sequence to

---

[1]In a valid alignment, at each step $u$ the output character is either $y_u$ or $\varnothing$ so that the concatenation of the tokens matches $\boldsymbol{y}$ after ignoring the $\varnothing$.

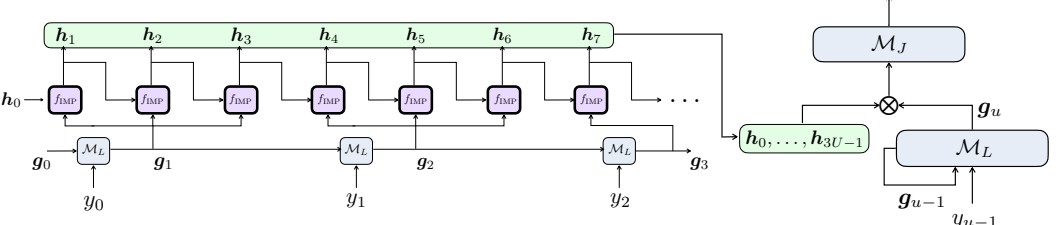

Figure 3: RNN-T finetuning using imputed speech representations from $f_{\text{IMP}}$ with alignments from the FixedGram model.

sequence model that takes as input text tokens $y_1, \ldots y_U$ and generates $\boldsymbol{h}_1, \ldots, \boldsymbol{h}_T$ using a standard encoder-decoder network. However, such an approach would require training a heavy-weight model from scratch. Instead, we developed a simple low-overhead model that we present next.

**Designing the Low-overhead Imputation Model**    Our approach is to leverage the existing trained RNN-T model $\mathcal{M}$ to reduce the overheads of generating the $\boldsymbol{h}$ sequence given a token sequence $\boldsymbol{y}$. First, we reduce the need for encoding discrete tokens $y_1, \ldots, y_U$ from scratch by starting from the output $\boldsymbol{g}_1, \ldots, \boldsymbol{g}_U$ of the language model $\mathcal{M}_L$. Second, we reduce the need for cross-attention training of a full seq2seq approach by pre-aligning the $\boldsymbol{h}$ sequence to the token sequence $\boldsymbol{y}$. We denote an alignment as $A$, a sequence of $(t, u)$ pairs to show that the LM state $\boldsymbol{g}_u$ was aligned with audio frame $\boldsymbol{h}_t$ in a valid execution of the RNN-T to generate a sequence. We will discuss how we generate such alignments during training and deployment of the Imputation model. Given an alignment $A$ we factorize the generator of the $\boldsymbol{h}$ sequence as follows:

$$\mathcal{P}(\boldsymbol{h}_1, \ldots, \boldsymbol{h}_T | y_1, \ldots, y_U, A) = \prod_{(t,u) \in A} P_h(\boldsymbol{h}_t | \boldsymbol{h}_{t-1}, \boldsymbol{g}_u) \tag{1}$$

In the above we have further assumed that $\boldsymbol{h}_t$ depends only on $\boldsymbol{h}_{t-1}$ and is independent of other prior $\boldsymbol{h}_r$s given $\boldsymbol{h}_{t-1}$. We model the above distribution using an imputation model $f_{\text{IMP}}$ which consists of a simple feedforward-neural networks that takes as input the language model encoding $\boldsymbol{g}_u$ and the last audio encoding $\boldsymbol{h}_{t-1}$ to produce the next audio encoding $\boldsymbol{h}_t$. Figure 2 presents an overview.

**Training the Imputation Model**    To create the training data for the imputation model $f_{\text{IMP}}$, we use the training data $D$ of the ASR model. For each utterance $\boldsymbol{x}^i$ in $D$, we trace the lattice of the highest probability beam to find the best valid alignment $A^i$ that consists of a sequence of $\boldsymbol{h}_t$ and the aligned $\boldsymbol{g}_u$ as shown in Figure 2. We use the alignment-length synchronous decoding algorithm (Saon et al., 2020) for this. From these alignments we extract training instances for the imputation model as $\{(\boldsymbol{h}_{t-1}, \boldsymbol{g}_u), \boldsymbol{h}_t\}$ for each $(t, u) \in A^i$. When multiple $\boldsymbol{g}_u$'s align with the same $\boldsymbol{h}_t$ in the best alignment output, we select the last $\boldsymbol{g}_u$. We train the parameters of the $f_{\text{IMP}}$ model to minimize the reconstruction loss of $\boldsymbol{h}_t$ with input $(\boldsymbol{h}_{t-1}, \boldsymbol{g}_u)$ as follows:

$$\hat{f_{\text{IMP}}} = \arg\min_f \sum_{i \in D} \sum_{(t,u) \in A^i} \text{Loss}(\boldsymbol{h}_t^i, f(\boldsymbol{h}_{t-1}^i, \boldsymbol{g}_u^i)) \tag{2}$$

where $\boldsymbol{h}_0 = 0$ and the space of $f$ are parameters of a standard feed forward network as described above. For the loss function, we considered several candidates and found the L1 distance to provide the best results as we show in our ablation studies.

## 3.2    RNN-T Finetuning using the Imputation Model

In this section we describe how we fine-tune the RNN-T model $\mathcal{M}$ on the target text $\tilde{D}$ using the trained $f_{\text{IMP}}$ model. For each text $\boldsymbol{y}^i \in \tilde{D}$ we first sample an alignment $A^i$ as elaborated in Section 3.2.1. Next use $A^i$ and $f_{\text{IMP}}$ to sample a sequence of vectors $\boldsymbol{h}^i : \boldsymbol{h}_1^i, \ldots, \boldsymbol{h}_{|A^i|}^i$ as shown in the alignment example in Figure 2. We first feed the token sequence $\boldsymbol{y}$ to the language module to get a sequence of outputs $\boldsymbol{g}_1, \ldots, \boldsymbol{g}_U$. Next we invoke the imputation model $f_{\text{IMP}}$ to generate for each $(t, u) \in A$ in increasing order of $t$, $\boldsymbol{h}_t \sim f_{\text{IMP}}(\boldsymbol{h}_{t-1}, \boldsymbol{g}_u)$. This gives us a pseudo labeled training set $\tilde{\mathcal{D}}_{\text{Imp}} = \{(\boldsymbol{h}^i, \boldsymbol{y}^i)\}$ that we use to fine-tune parameters $\boldsymbol{\theta}_L$ and $\boldsymbol{\theta}_J$ of the RNN-T using the same loss

function of maximizing the likelihood of $\boldsymbol{y}$ over all possible alignments. Since the generated $\boldsymbol{h}$ come from the $\Pr(\boldsymbol{h}|\boldsymbol{y})$ distribution, and only the $\mathcal{P}(\boldsymbol{y})$ distribution changes in the target distribution, this fine-tuning step adapts to the target distribution.

$$\min_{\boldsymbol{\theta}_L, \boldsymbol{\theta}_J} \sum_{(\boldsymbol{h}, \boldsymbol{y}) \in \tilde{\mathcal{D}}_{\text{Imp}}} \text{RNNT-loss}(\boldsymbol{y}, \{\mathcal{M}_J(\cdot | \boldsymbol{g}_u = \mathcal{M}_L(y_1, \ldots, y_{u-1}), \boldsymbol{h}_t)\}) \quad (3)$$

### 3.2.1 GENERATING ALIGNMENTS GIVEN A TEXT SEQUENCE

We tried many different light-weight methods of generating alignments for a given text token sequence $\boldsymbol{y}$. A simple method is to generate a fixed number of blanks($\varnothing$) B before each token $y_u$. We call this the FixedGram model. A second option is to train a distribution over the number of blanks for each token $v \in \mathcal{V}$. Finally, a more evolved method would be to use a state-based model, much like the imputation model that takes as input $g_u$ and $h_{t-1}$ to generate if the next token should be a blank. We will show that the simple FixedGram model was adequate for the adaptation task. Figure 3 presents an overview of our steps during fine-tuning under the FixedGram alignment model. Algorithm 1 presents the overall pseudocode of TOLSTOI. Note, the training of the imputation model is performed once and is independent of the adaptation dataset.

---

**Algorithm 1** Text-only adaptation in TOLSTOI

**Require:** Trained RNN-T model $\mathcal{M}$, Training data $D$, Adaptation Text $\tilde{D}$
  **for** $(\boldsymbol{x}^i, \boldsymbol{y}^i) \in D$ **do**
    $A^i, \boldsymbol{h}^i, \boldsymbol{g}^i$ = Apply $\mathcal{M}$ on speech $\boldsymbol{x}$ to extract alignments and $\boldsymbol{h}, \boldsymbol{g}$ vector sequences
  **end for**
  $f_{\text{IMP}}$ = Train imputation model on aligned pairs $\boldsymbol{h}_t^i, \boldsymbol{g}_u^i$ using Equation 2
  /* Fine-tune $\mathcal{M}$ for $\tilde{D}$ */
  **for** $\boldsymbol{y} \in \tilde{\mathcal{D}}$ **do**
    $\boldsymbol{g}_1 \ldots \boldsymbol{g}_U = \mathcal{M}_L(\boldsymbol{y}), \quad \boldsymbol{h}_0 = 0$
    $A$ = GenerateAlignments($\boldsymbol{g}$)
    **for** $(t, u) \in A$ in increasing order of $t$ **do**
      $\boldsymbol{h}_t = f_{\text{IMP}}(\boldsymbol{h}_{t-1}, \boldsymbol{g}_u)$
    **end for**
    loss = RNN-T_loss(softmax($\mathcal{M}_J(\boldsymbol{h}, \boldsymbol{g})$), $\boldsymbol{y}$)
    Backpropagate(loss, $\{\boldsymbol{\theta}_J, \boldsymbol{\theta}_L\}$)
  **end for**
  Return $\boldsymbol{\theta}_L, \boldsymbol{\theta}_J$

---

## 4 EXPERIMENTS

We present an extensive evaluation of TOLSTOI against three existing approaches with two ASR models and three target domains.

### 4.1 EXPERIMENTAL SETUP

We perform adaptation experiments on two RNN-T models of very different capacity: (i) **SWB 300H** - trained on the Switchboard 300H dataset (Godfrey et al., 1992) containing over 300 hours of labelled training data and (ii) **SWB 2000H** - trained on 262 hours of Switchboard-300, 1698 hours of Fisher data (Cieri et al., 2004) and 15 hours of CallHome data (Martin & Przybocki, 2000).

The transcription network of the RNN-T model consists of 6 bidirectional LSTM layers with a hidden size of 640 cells and a projection layer that projects down the encoder embeddings to 256 dimensions. The prediction network consists of a single-layer unidirectional LSTM with a hidden size of 768 cells and a projection layer reducing the prediction network embeddings to 256 dimensions. The joint network first combines the transcription network embeddings (256-d) and prediction network embeddings (256-d) via a Hadamard product joint operation ($\odot$) (Saon et al., 2021), followed by a tanh non-linearity and a final output softmax layer that produces a probability distribution over 45 non-blank characters ($\mathcal{V}$) and a blank character ($\varnothing$). The total number of parameters in the RNN-T model is roughly 56 million.

The audio features are mean and variance normalized log-Mel filterbank features computed for every 10ms of the audio. These features are further appended with delta-spectral and double-delta spectral features( (Mason & Zhang, 1991; Picone, 1993)) where every two consecutive frames are stacked resulting in 240-dimensional (240-d) input vectors with a receptive field of 20ms. Speed and tempo perturbations are applied to each input with the value of $0.9, 1.1$ to augment the training data. We also use SpecAugment (Park et al., 2019) for additional augmentation. The RNN-T models were trained for 20 epochs using AdamW (Loshchilov & Hutter, 2017) optimizer with the maximum learning rate of 5e-4 and the OneCycleLR policy (Smith & Topin, 2019) policy consisting of a linear warmup phase from 5e-5 to 5e-4 followed by a linear annealing phase to 0. A batch size of 64 was used to train the Pytorch models on V100 GPUs.

**Adaptation Datasets** For the adaptation experiments, we choose three diverse datasets with different amounts of adaptation text coming from three different domains. (1) **ATIS** (Hemphill et al., 1990) consists of roughly 5K sentences from the airline reservation domain for training and 893 (speech, text) utterance pairs for testing. (2) **HarperValleyBank (HVB)** (Wu et al., 2020) consists of roughly 15K sentences from the banking domain for training and 2797 (speech, text) utterance pairs for testing. (3) **Librispeech** (Panayotov et al., 2015) consists of roughly 29K sentences from audiobooks for training and 2619 (speech, text) utterances for testing. Note, in true spirit of text-only adaptation, we do not assume the availability of a parallel validation dataset in the target domain.

**Imputation Model** The imputation model of TOLSTOI is a 2-layer feed-forward network with a tanh non-linearity. The first layer projects the 512-d input of $(h_{t-1}, g_u)$ to 256-d and the second layer outputs a 256-d $h_t$. The imputation model contains roughly 200K parameters, in contrast to the 56 million parameters of the ASR model. The imputation model is trained using the same learning rate, optimizer, and learning rate schedule as the baseline training with a batch size of 2048. The SWB 300H training data was aligned with the corresponding ASR model yielding a total of 42 million $(h_t, g_u)$ pairs to train the imputation model.

**Fine-tuning details** For fine-tuning of $\mathcal{M}_L$ using the imputation model, we keep the same optimizer and learning rate scheduler as the starting RNN-T training except that the maximum learning rate used for fine-tuning was 5e-5. We fine-tune for a fixed number of 2000 updates since we do not have a validation set for target-specific hyper-parameter selection.

**Metrics** We evaluate on three metrics: (1) **RTF**: Real Time Factor(RTF) which measures the average time (in seconds) to decode one second of audio (50 audio frames in our case). It is computed by averaging over the entire test set on a single CPU. (2) **WER on Target-Only**: Word error rate(WER) on the test set of the target domain. (3) **WER on Source-Target Mixture**: The test sizes for each of the target domains datasets is small. In any ASR system deployed in the real world, it is important for the adapted model to not be significantly worse than the unadapted model on sentences outside its narrow domain because of catastrophic forgetting during adaptation. Therefore, we also measure the average WER of the target and source test datasets. The test data of the source domain comprises 4458 (speech utterance, text) pairs from Hub5'00 and CallHome.

We compare TOLSTOI against three existing methods of text-only adaptation of RNN-Ts.

**Shallow Fusion** (Kannan et al., 2018) is a standard method which uses an external LM trained on the target domain text to interpolate the RNN-T probabilities during inference. For the external LM, we use the same configuration as our prediction network and set the interpolation parameter ($\lambda$) to 0.3 for all cases. Since we do not assume access to any validation set from the target domain, fine-tuning the hyper-parameter is not an option.

**NN-LM** In this method, the prediction network is viewed as an auto-regressive language model. To get the LM scores, a linear layer is added to project the prediction network representations to the output vocabulary space $\mathcal{V}$ and fine-tuned on the target domain text for one epoch.

**Textogram** (Thomas et al., 2022) In this method, the transcription network is modified to work with two input modalities: (i) standard acoustic features and (ii) one-hot encoding of the units in the text, with the encoding of each unit being repeated a fixed number of times such that each feature has duration similar to the spectrograms. During the training of the base model, either the acoustic feature or textogram feature is used for training the RNN-T objective. During fine-tuning, only the textogram features for each text from the target domain are used for adaptation. Different from NN-LM, in textogram RNN-T loss is used for fine-tuning the joint network and prediction network parameters much like in our method.

**Other methods** We also compared with two other methods: (1) The factorized model proposed in Chen et al. (2022a). However, their factorized RNN-T model had significantly worse accuracy than our unadapted RNN-T model both before and after text-only adaptation. (2) TTS where we used a proprietary IBM synthesis engine (Kons et al., 2019; Fernandez et al., 2022) based on Tacotron (Shen et al., 2018) to generate audio for the text-only inputs. This gives competitive reduction in WERs of target-only/source-target mixed (3.2/6.5 and 8.6/9.8 on the ATIS and HVB test sets, respectively). However, it is worth noting here that all three target domains consist of US-accented English speech and the TTS system we employ is carefully tuned to produce natural-sounding US-accented English samples. Building TTS of similarly high quality for low-resource languages is a non-trivial

| Baseline Model | Method | RTF↓ (sec) | ATIS (WER(↓)) | | HVB (WER(↓)) | | Librispeech (WER(↓)) | |
|---|---|---|---|---|---|---|---|---|
| | | | Target | Mixed | Target | Mixed | Target | Mixed |
| SWB 2000H | Unadapted | 0.33 | 5.8 | 7.3 | 17.5 | 13.1 | 12.5 | 10.6 |
| | NN-LM | 0.33 | 5.7 | 7.2 | 16.4 | 13.1 | 12.1 | 10.5 |
| | Textogram | 0.72 | 5.4 | 19.2 | 12.2 | 26.0 | 14.1 | 23.6 |
| | Shallow Fusion | 0.85 | 4.3 | 15.0 | 12.8 | 11.9 | 12.0 | 10.8 |
| | TOLSTOI | 0.33 | 4.0 | **6.6** | 11.2 | **10.4** | 11.1 | **10.0** |
| SWB 300H | Unadapted | 0.33 | 12.5 | 12.6 | 34.4 | 23.5 | 20.3 | 16.5 |
| | NN-LM | 0.33 | 12.4 | 12.6 | 33.3 | 23.6 | 20.3 | 16.5 |
| | Textogram | 0.72 | 10.8 | 22.3 | 24.5 | 33.6 | 19.1 | 19.2 |
| | Shallow Fusion | 0.85 | 8.1 | 13.7 | 29.0 | 23.8 | 19.8 | 17.2 |
| | TOLSTOI | 0.33 | 10.4 | **12.0** | 23.8 | **18.8** | 19.1 | **16.1** |

Table 1: Comparison on decode time (RTF) and WERs on target-only and target-source mixed data of different adaptation methods on two different ASR models trained on SWB 2000H (top) and SWB 300H (bottom) adapted to three different domains. WERs on SWB test are 8.8 and 12.7, respectively, using SWB-2000 and SWB-300. When deployed on an equal mixture of source and target data, TOLSTOI provides the highest reduction in WER.

| Reference | i need to reset i would like to reset my password |
|---|---|
| Shallow Fusion | i need to reset i would like to reset my password |
| TOLSTOI | i need to reset i would like to reset my password |
| Reference | hi i lost my debit card can you send me a new one my name is robert davis by the way |
| Shallow Fusion | hi i lost my debit card can you send me a new one my name is robert davis by the way |
| TOLSTOI | hi i lost my debit card can you send me a new one my name is robert davis by the way |

Table 2: Anecdotal examples comparing TOLSTOI with shallow fusion. Deletion errors are underlined and highlighted in red.

task (Ogayo et al., 2022). In contrast, TOLSTOI relies only on access to the ASR pretraining data to train the imputation model and can potentially scale well to low-resource languages.

## 4.2 OVERALL COMPARISONS

Table 1 presents the decode time (RTF) and target-only WER and source-target mixed WER of different adaptation methods on two ASR models adapted to three different target domains. If we focus only on target WERs, shallow fusion provides good reductions in WER. However, this comes at the cost of a 2.5 times blow-up of decode time (RTF) which could be unacceptable in many applications. Also, the external LM fine-tuned on the target text leads to significant catastrophic forgetting as observed by the significantly increased WER on the mixed corpus. TOLSTOI provides the best WER when measured on the mixed test set while being competitive with shallow fusion on the target-only set. The decode time of TOLSTOI is exactly the same as the basic RNN-T since we do not modify the RNN-T architecture. NN-LM does not provide much gains beyond the unadapted model. The Textogram is more successful in adaptation but is also subject to significant deterioration on the mixed corpus. Also, the Textogram modified the network architecture of the RNN-T resulting in significantly larger decode time compared to the basic RNN-T.

Table 2 presents two illustrative examples from ATIS comparing predictions using shallow fusion and TOLSTOI. Shallow fusion does not handle disfluencies well if the target text used to train the external LM is devoid of it. For example, the false start "I need to reset" and the filler phrase "by the way" are omitted. More anecdotal examples are shown in Table 7 in Appendix A.

## 4.3 ABLATION STUDY

Table 3 presents ablations of various design choices governing TOLSTOI. We investigate three dimensions of the imputation model: 1) choice of loss function, 2) choice of alignment to sample at test-time, and 3) choice of model architecture. On the choice of loss function, we replaced the L1 loss in Eqn 2 with an L2 loss and a contrastive loss. We used the contrastive loss from Chen et al. (2020) for the imputed features in each batch. We also tried a distillation loss where a generated $h_t$ is combined with the corresponding $g_u$ via $\mathcal{M}_J$ to generate a probability distribution over the vocabulary and a KL-Divergence loss is imposed between this predicted distribution and the probability distribution from the real $h_t$. These losses were all worse than L1. On choice of the alignment

| Method | ATIS | | HarperValleyBank | |
|---|---|---|---|---|
| | Target | Mixed | Target | Mixed |
| Unadapted | 5.8 | 7.3 | 17.5 | 13.1 |
| TOLSTOI | 4.0 | 6.6 | 11.2 | 10.4 |
| **Imputation Model Loss Function** | | | | |
| Imputation Model with L2 Loss | 4.2 | 6.5 | 11.3 | 10.4 |
| Imputation Model with Contrastive Loss | 7.8 | 8.6 | 17.1 | 13.1 |
| Imputation Model with Distillation Loss | 5.3 | 7.1 | 15.4 | 12.4 |
| **Alignment Generation Models** | | | | |
| Imputation Model with 1 blank | 7.2 | 8.1 | 23.4 | 16.4 |
| Imputation Model with 4 blanks | 4.2 | 6.6 | 11.3 | 10.4 |
| Imputation Model with dynamic length model | 4.4 | 6.7 | 11.3 | 10.4 |
| **Imputation Model Architecture** | | | | |
| $(\boldsymbol{g}_u)$ | 9.1 | 9.0 | 9.5 | 16.0 |
| $(\boldsymbol{h}_{t-1}, \boldsymbol{g}_u, \boldsymbol{g}_{u+1})$ | 4.4 | 6.7 | 11.8 | 10.7 |
| $(\boldsymbol{h}_{t-1}, y_{u-1}, y_u, y_{u+1})$ | 5.6 | 7.4 | 15.6 | 12.5 |
| Transformer Imputation Model | 5.7 | 7.2 | 17.3 | 13.0 |
| - No Switchboard Data | 4.4 | 7.2 | 11.7 | 11.6 |

Table 3: Ablation of various design choices of TOLSTOI. More complicated loss functions, or alignment models, or imputation models performed worse than the simple design choices of TOLSTOI.

generation model, we tried the FixedGram model with 1 blank or 4 blanks before each output token instead of the default of 3. We also tested a feed-forward regression model for length that takes $g_u$ as its input and outputs the number of speech frames learnt from the alignment data (c.f., Section 3.1). All these alignment generation models fared worse than using a FixedGram model with 3 blanks as in TOLSTOI. We next tried different tweaks to the architecture of the imputation model. We tried changing the input context to the imputation model by adding $\boldsymbol{g}_{u+1}$, adding output characters $y_{u-1}, y_u, y_{u+1}$ or omitting $\boldsymbol{h}_{t-1}$. We also tried a Transformer-based (Vaswani et al., 2017) encoder-decoder imputation model to transform the complete $\boldsymbol{g}_u$ sequence to a full $\boldsymbol{h}_t$ sequence. All these variants performed worse than TOLSTOI. During RNN-T finetuning, we found it useful to mix an equal number of utterances from the pretraining corpus with the imputed data (Zhu et al., 2022). Omitting this step resulted in small WER degradations shown in the last row in Table 3.

Figure 4 shows the WERs on ATIS and HVB as a function of decreasing amount of target text. As expected, there is an increase in WERs with reducing the amount of target text. However, the reduction in WERs with using only 50% of the data versus using the complete dataset is not substantial. This suggests that TOLSTOI could work well even in target domains with very limited amounts of text-only data.

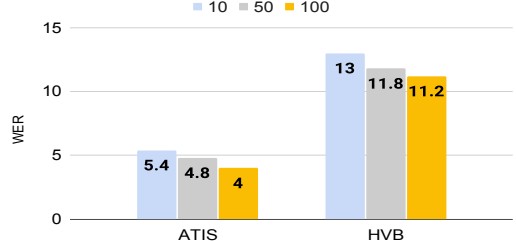

Figure 4: WER vs. Adaptation Text(%)

## 5 CONCLUSION

In this paper we presented TOLSTOI, a low-overhead method of adapting RNN-T models using text-only data in a target domain. Our key insight it to impute speech features from the last layer of the transcription network, which allows accurate fine-tuning of a subset of the RNN-T parameters. We proposed a very simple design of the imputation model by leveraging existing text-speech representations and alignments from the trained RNN-T model. Unlike existing methods, TOLSTOI does not modify the base RNN-T architecture and can adapt existing pre-trained ASR models without increasing inference time during decoding. Via experiments on three target domains and two ASR models, we show that TOLSTOI provides the best accuracy on a mixed source-target test set since it is least subject to catastrophic forgetting while reducing target WER by more than 35%. With a detailed ablation over more complicated models, we justify the effectiveness of our simple imputation model. As part of future work, we would like to experiment our approach on low-resource languages and explore techniques for online adaptation of the ASR model.

## 6 REPRODUCIBILITY STATEMENT

All our experiments are performed on publicly available datasets such as Switchboard, ATIS, HarperValleyBank, and Librispeech. We also use published train/test splits specified for each of these datasets, thus enabling reproducibility. We have provided sufficient implementation details of our baseline models, our imputation model and the RNN-T fine-tuning process to help reproduce our main results.

## 7 ACKNOWLEDGEMENTS

We would like to thank George Saon, Samuel Thomas and Jeff Kuo for insightful discussions. The authors from IIT Bombay gratefully acknowledge support from IBM Research, specifically the IBM AI Horizon Networks-IIT Bombay initiative.

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

## A  APPENDIX

### A.1  EVALUATION OF TOLSTOI USING A CONFORMER MODEL

In this section, we compare the effectiveness of TOLSTOI using a Conformer (Gulati et al., 2020) encoder in the RNN-T. While FixedGram-based TOLSTOI works well on the bidirectional-LSTM encoder as shown in Section 4.2, the purpose of this experiment is to evaluate its effectiveness for the Conformer encoder which would distribute acoustic features in the neighborhood. The Conformer-Transducer model uses an encoder network of 10 Conformer blocks (512-dimensional feed-forward module, 31-kernel convolution block, and 8 64-dimensional attention heads). We train the model on the SWB 300H dataset for this experiment. All the other model architecture choices and hyperparameters remain the same.

| Method | ATIS | | HVB | |
|---|---|---|---|---|
| | Target | Mixed | Target | Mixed |
| Unadapted | 11.5 | 11.9 | 39.7 | 26.0 |
| NN-LM | 11.3 | 11.9 | 38.1 | 25.6 |
| Shallow Fusion | 9.1 | 14.1 | 32.2 | 23.6 |
| TOLSTOI | 10.4 | 12.4 | 28.8 | 21.9 |

Table 4: Comparison on WERs on target-only and target-source mixed data of different adaptation methods on conformer-based RNN-T ASR model trained on SWB 300H adapted to ATIS and HVB domains.

We empirically observe that the FixedGram-based TOLSTOI works well with the Conformer-based RNN-T model. The reduction in the WER for the target domains is significant and catastrophic forgetting is also minimal compared to shallow fusion.

### A.2  EVALUATION OF TOLSTOI ON RNN-T MODEL WITH BPE UNITS

In this section, we compare the effectiveness of TOLSTOI on an RNN-T model trained on subword-based BPE units (Sennrich et al., 2015). While the FixedGram-based TOLSTOI works well on the bidirectional-LSTM encoder with character units, the purpose of this experiment is to evaluate the effectiveness of the model with subword units. To this end, we train an SWB 300H model using the subword units using the same model configuration as for the character model, except that our output vocabulary now comprises 1000 BPE units as opposed to the 45 characters. The subword encoder is learned on the text obtained from the training set. We show results on TOLSTOI using the FixedGram approach.

| Method | ATIS | | HVB | |
|---|---|---|---|---|
| | Target | Mixed | Target | Mixed |
| Unadapted | 12.7 | 12.5 | 41.9 | 27.1 |
| NN-LM | 12.5 | 12.5 | 39.8 | 26.2 |
| Shallow Fusion | 11.1 | 22.6 | 34.2 | 35.0 |
| TOLSTOI | 10.0 | 11.5 | 33.1 | 22.9 |

Table 5: Comparison on WERs on target-only and target-source mixed data of different adaptation methods on subword-based RNN-T model trained on SWB 300H adapted to ATIS and HVB domains.

We empirically observe that the FixedGram-based TOLSTOI works well even with the subword units. The reduction in the WER for the target domains is significant and catastrophic forgetting is also minimal as compared to the Shallow Fusion. Additionally, we observe that the catastrophic forgetting for shallow fusion increases for the BPE-based models as opposed to the character-based model.

### A.3 Evaluation of TOLSTOI on RNN-T model trained on extremely large dataset

In this section, we compare the effectiveness of the TOLSTOI on an RNN-T model trained on an extremely large dataset consisting of proprietary data totaling close to 56K hours of labeled speech.

| Method | ATIS | | HVB | |
|---|---|---|---|---|
| | Target | Mixed | Target | Mixed |
| Unadapted | 5.2 | 6.8 | 12.2 | 10.3 |
| NN-LM | 5.1 | 6.8 | 12.0 | 10.3 |
| Shallow Fusion | 2.8 | 7.1 | 8.6 | 12.2 |
| TOLSTOI | 2.8 | 5.9 | 7.9 | 8.7 |

Table 6: Comparison on WERs on target-only and target-source mixed data of different adaptation methods on RNN-T model trained on proprietary data adapted to ATIS and HVB domains.

We empirically observe that the FixedGram-based TOLSTOI works well even when used with very large amounts of training data. Our results at such a scale are consistent with our results on smaller SWB 300H and SWB 2000H benchmarks.

### A.4 Anecdotal Examples

| | |
|---|---|
| Reference | we are going to fly listen we are going to fly over saint louis |
| Unadapted | we are going to fly listen we are going to fly over saint louis |
| Shallow Fusion | we are going to fly listen we are going to fly over saint louis |
| TOLSTOI | we are going to fly listen we are going to fly over saint louis |
| Reference | he went moved to texas |
| Unadapted | he went moved to texas |
| Shallow Fusion | he went and moved to texas |
| TOLSTOI | he went moved to texas |
| Reference | i would like to transfer money between my accounts |
| Unadapted | i would like to transfer money between my accountants |
| Shallow Fusion | i would like to transfer money between my account |
| TOLSTOI | i would like to transfer money between my accounts |
| Reference | yeah we played softball last night we were clobbered |
| Unadapted | yeah we played softball last night we were clobbered |
| Shallow Fusion | yeah we played softball last night we were clovered |
| TOLSTOI | yeah we played softball last night we were clobbered |

Table 7: Anecdotal examples on the mixed test set comparing TOLSTOI with shallow fusion. Deletion errors are underlined and highlighted in red.

| Reference | thanks you too |
|---|---|
| Unadapted | thanks you too |
| Shallow Fusion | thanks you too |
| TOLSTOI | thank you TODAY |
| Reference | they plan to get married in topeka since most of their friends are there |
| Unadapted | they plan to get married in to figure since most of their friends are there |
| Shallow Fusion | they plan to get married into since most of their friends are there |
| TOLSTOI | they plan to get married in to peak us since most of their friends are there |
| Reference | show me weekday flights from milwaukee to orlando one way |
| Unadapted | show me weak day flights from milwaukee to orlando one way |
| Shallow Fusion | show me weekday flights from milwaukee to orlando one way |
| TOLSTOI | show me weakday flights from milwaukee to orlando one way |

Table 8: Anecdotal examples showing the mistakes with TOLSTOI and shallow fusion. Errors are underlined and highlighted in red. When the language model context is not clear (e.g, for word topeka), TOLSTOI produces acoustically similar predictions as opposed to the Shallow Fusion and Unadapted models.

