# OpenReview forum: "In-Situ Text-Only Adaptation of Speech Models with Low-Overhead Speech Imputations"
_ICLR.cc/2023/Conference — ICLR 2023 poster_

### Official Review · Reviewer_USzV · 2022-10-24

**Confidence:** 4
**Correctness:** 3
**Technical Novelty And Significance:** 3
**Empirical Novelty And Significance:** 2
**Recommendation:** 6

**Clarity, Quality, Novelty And Reproducibility:**

 Quality and presentation are good. Details are provided and hence it would not be hard to reproduce. However, similar ideas have been presented in prior studies.

**Strength And Weaknesses:**

Strengths
* The design of the imputation model is well motivated (keeping inference time short, drop-in replacement for existing models, good performance on both target and general domain)
* It is very easy to follow the paper and all details are included.
* Performance is strong compared to the baselines. Ablation studies are informative, showing a) the impact of loss function, b) simple alignment generation is sufficient, c) reusing $M_L$ embedding simplifies the imputation model.

Weakness
* The idea of generating internal representations of an end-to-end model to fine-tune a subset of parameters has been studied before [1,2]. I can see that the design of the “imputation models” (or so-called text-to-embedding (TTE) models in prior studies) are different. However, the authors should discuss these studies and present experiments to compare with prior models.

[1] Hayashi, Tomoki, et al. "Back-translation-style data augmentation for end-to-end ASR." 2018 IEEE Spoken Language Technology Workshop (SLT). IEEE, 2018.

[2] Hori, Takaaki, et al. "Cycle-consistency training for end-to-end speech recognition." ICASSP 2019-2019 IEEE International Conference on Acoustics, Speech and Signal Processing (ICASSP). IEEE, 2019.


**Summary Of The Paper:**

This paper studies RNN-T ASR model adaptation using unpaired text in the target domain. The authors proposed to use a module called “Imputation Model” which predicts speech representation of the speech encoder ($M_s$) given text, such that text encoder $M_L$ and the joint network $M_j$ in RNN-T can be fine-tuned using text and pseudo speech embedding.

The use of the imputation model is compatible with any existing RNN-T model, does not increase the total number parameters during inference, suffer milder catastrophic forgetting, and yields better performance compared to the included baselines (shallow fusion, $M_L$ as LM, textogram).


**Summary Of The Review:**

This paper presents a practical solution to RNN-T ASR model adaptation using unpaired text. Experiments and presentations are good. I would encourage the author to improve the comparison with existing works.

---

> ### Author Response · Authors · 2022-11-19
> **Author's response to reviewer USzV**
>
> We thank the reviewer for their insightful comments and for pointing out relevant related work.
>
> We compare against the numbers reported in [1] and [2] on the test-clean Librispeech test-set using 100 hours of paired Librispeech training data and 360 hours of text-only (Librispeech) data. We train our imputation model using 100 hours of paired Librispeech and generate representations for the text-only inputs that are further used to finetune the ASR model.
>
> ***
>                               | Back-Translation style     | Cycle-Consistency |  TOLSTOI  |
>                               | Augmentation [1]           | Training [2]      |           |
> ***
>
>     Baseline                  |  25.2                      |    25.2           |  14.4     |
>     Finetuned                 |  23.6                      |    21.5           |  13.3     |
>     Relative WER Reduction    |  6.3%                      |    14.7%          |  7.6%     |
> ***
>
>
> Since the reported baseline numbers in [1] and [2] are much worse than our baseline, we only focus on relative WER reductions. We observe similar reductions as in [1]. While [2] reports larger relative WER reductions, we note that [2] is not comparable to our method since they use unpaired audio from the 360 hour subset to train the ASR model with a cycle-consistency loss. [2] did not report a text-only experiment (as in [1]); that would have been a fairer comparison to TOLSTOI.
>
> > References:
>
> [1] Hayashi, Tomoki, et al. "Back-translation-style data augmentation for end-to-end ASR." 2018 IEEE Spoken Language Technology Workshop (SLT). IEEE, 2018.
>
> [2] Hori, Takaaki, et al. "Cycle-consistency training for end-to-end speech recognition." ICASSP 2019-2019 IEEE International Conference on Acoustics, Speech and Signal Processing (ICASSP). IEEE, 2019.
>
> [3] Shen, Jonathan, et al. "Natural tts synthesis by conditioning wavenet on mel spectrogram predictions." 2018 IEEE international conference on acoustics, speech and signal processing (ICASSP). IEEE, 2018.

---

> > ### Comment · Reviewer_USzV · 2022-12-11
> > **thanks for the additional comparison**
> >
> > the additional results have addressed my question. i'm raising my rating to 6

---

### Official Review · Reviewer_cvPh · 2022-10-24

**Confidence:** 4
**Correctness:** 4
**Technical Novelty And Significance:** 4
**Empirical Novelty And Significance:** 3
**Recommendation:** 6

**Clarity, Quality, Novelty And Reproducibility:**

The paper is very well written, easy to follow and to read and describe a compelling idea. It is well structured and contains the right level of details.

The idea seem novel, although it could be seen as an evolution of the textogram method or something between TTS generation and textogram. It addresses a very relevant problem for end-to-end ASR and its adaptability to real-life industrial scenarios.

It contains a lot of details that should help reproduce the results, as stated by the authors, although making the code or a recipe available would be even better.

**Strength And Weaknesses:**

Strength: the proposed method is very interesting, does not require to retrain the whole network, gives good results, and seems quite simple to implement and to test. Among the different design choices, the simplest ones (L1 loss, fixed gram for alignment generation, imputation network architecture) seem to work best, which is also a nice result. Finally, a pretty small imputation network looks sufficient.

Weakness: The evaluated RNN-Ts predict characters, while is looks more common to use subwords. It would be nice to see if the results tranfer also to subword modeling. Although the focus here is not really rare words or named entities, I would also be curious to see the results for this challenging task, compared to biasing techniques.


**Summary Of The Paper:**

In this paper, the authors propose a text-only adaptation method for RNN-Ts, allowing to make these models better for a new domain than the one represented in the training data. While existing methods require a change in the RNN-T architecture, a full retraining, or introduce some latency, the proposed approach allows to only fine-tune the language and joint network, similarly to methods relying on TTS. Here, instead of generating audio for domain-specific texts using TTS, the output representations of the speech model are generated from the text directly.

The model that generates these intermediate representations is trained using the forced alignments of the labeled training dataset. A small recurrent neural network is trained to predict the next hidden speech vector from the current one and current language hidden vector. To adapt to a new domain, alignments are first generated from the text, and the small network is used to predict a sequence of hidden speech vectors. These vectors are in turn used to fine-tune the language and joint networks.

Through experiments with two base training sets and 3 adaptation domains, the paper shows that the proposed method with a 56M-parameter model outperforms the existing adaptation approaches without a significant increase of latency or degradation of performance on the original domain.

**Summary Of The Review:**

The paper is good and the idea worth sharing with the community. The ability to adapt an end-to-end ASR model is important and represents a challenging task, and the presented method allows to do so without changing the RNN-T architecture and without requiring a complete fine-tuning of the model, which not only saves energy but also allows to use that method without requiring a lot of computational power.

For these reasons, I think the paper is interesting for ICLR.

Some clarifications and additional experiments could nevertheless add more value to this paper:

   - is teacher forcing used for training imputation model?
   - to measure catastrophic forgetting, WER is computed on a source-target mixture dataset: why not measure on source and target separately?
   - is the alignment sampling method (and the overall adaptation technique) also good when the model outputs subwords instead of characters?
   - the "Real Time Factor(RTF) which measures the average time (in seconds) to decode 50 audio frames": 50 audio frames is indeed, in the described setup, 1 second of audio: it would be clearer and more consistent with the literature to define RTF as the total processing time divided by the actual audio duration maybe?
   - more analysis of the representations learned by the imputation model, and maybe a focus on rare words would be very interesting for an ICLR paper

---

> ### Author Response · Authors · 2022-11-19
> **Author's response to reviewer cvPH**
>
> We thank the reviewer for their insightful comments.
>
> **Q1: Evaluation of TOLSTOI using subword-based ASR models.**
>
> **A1**: We trained an SWB-300H model using 1000 BPE subword units as its vocabulary (as opposed to 45 characters used in our original experiments).
>
> ***
>                       |       ATIS (WER)       |       HVB (WER)         |
>                       | Target       | Mixed   |   Target    | Mixed     |
> ***
>
>     Unadapted         |  12.7        |   12.5  |   41.9      |  27.1     |
>     NN-LM             |  12.5        |   12.5  |   39.8      |  26.2     |
>     Shallow Fusion    |  11.1        |   22.6  |   34.2      |  35.0     |
>     TOLSTOI           |  10.0        |   11.5  |   33.1      |   22.9    |
> ***
>
> We observe that TOLSTOI performs the best with subword-based ASR models as well, yielding the lowest WERs on both the target test sets and the mixed test sets (the latter by a large margin). We have added these results in the appendix of the paper.
>
> **Q2: Is teacher forcing used to train the imputation model?**
>
> **A2**: We do not use teacher forcing. We use the best-decoded paths via beam search to derive the alignments.
>
> **Q3: To measure catastrophic forgetting, why not measure on the source and target separately?**
>
> **A3**: With measuring source and target WERs separately, it becomes difficult to identify which technique performs best overall. The mixed WER offers a single metric that helps identify which technique improves both adaptation to a target domain while maintaining performance on the source domains.
>
> **Q4: Clarify RTF definition**
>
> **A4**: We thank you for pointing this out. We have updated the paper to clarify the RTF computation.
>
> **Q5: Analysis of representations learned by the imputation model and focus on rare words.**
>
> **A5**: The [tsne plot](https://freeimage.host/i/tsne.H9yTJbp) at the anonymised URL, compares the 256-dimensional representations obtained from (0) TOLSTOI and (1) actual audio from the SWB dataset corresponding to the same underlying text. We observe that there is a reasonable amount of overlap across both kinds of representations even with using a very lightweight imputation model.
>
> Regarding rare words, we do observe instances of TOLSTOI producing acoustically consistent predictions for rare words when the other baselines do not. For example, as shown in Table 5 in Appendix A, TOLSTOI predicts “to peak us since” for “topeka since” which is acoustically much closer than “into since” predicted by Shallow Fusion.

---

### Official Review · Reviewer_BcMJ · 2022-10-25

**Confidence:** 4
**Correctness:** 3
**Technical Novelty And Significance:** 4
**Empirical Novelty And Significance:** 3
**Recommendation:** 8

**Clarity, Quality, Novelty And Reproducibility:**

The paper is well-written and easy to understand. The idea is novel, and I believe this work can motivate many others. The paper supports the claim with key experiments. It seems that the training and architecture details are explained sufficiently, but I am not fully sure if there are some missing details.

**Details Of Ethics Concerns:**

There is no specific ethical concern for this paper.

**Strength And Weaknesses:**

Strengths:

- The paper tackles important and practical problems with a simple approach. The motivation of the paper is meaningful, and the related works are studied well and clearly demonstrated in a general framework (Figure 1).
- The success of FixedGram is interesting that a fixed number of blanks are sufficient for pseudo-generating the speech features.
- Measuring not only WER but also RTF clearly shows the advantage of the method. Also, it is good to consider catastrophic forgetting.

Weaknesses:
- The model architectures are slightly old-fashioned. How about trying Transformer-based acoustic models instead of LSTM-based ones? Also, I guess that in the case of ContextNet or Conformer, convolution operations would distribute the acoustic features and FixedGram may not work well.
- As I understand, the imputation model is removed after the adaptation during inference. It would be better if authors investigate the difference between learned IMP output and actual output from speech module (for both source and target domains).
- Just a question; training takes multiple steps; is “end-to-end” an appropriate explanation? It seems that there are many manual steps, for example, extracting alignments and training different modules with different hyperparameters.

**Summary Of The Paper:**

This paper proposes a novel text-only adaptation method for RNNT-based ASR systems. Unlike previous approaches (shallow fusion, TTS, fine-tuning), the proposed method called TOLSTOI is lightweight with a cheap training cost. The key idea of TOLSTOI is to train an “imputation model" that mimics the speech module outputs from the language module outputs. The imputation model is trained first, and then the language and joint models are fine-tuned. Experimental results show that the proposed method achieves the lowest WER when the source and target domains are different.

**Summary Of The Review:**

Overall, the paper is novel and tackles an important problem. The strengths are strong, and the weaknesses are minor. Although the scalability to other models needs further validation, I recommend this paper be accepted.

---

> ### Author Response · Authors · 2022-11-19
> **Author's response to reviewer BcMJ**
>
> We thank the reviewer for their insightful comments.
>
> **Q1: How does TOLSTOI perform with a more state-of-the-art Conformer architecture?**
>
> **A1**: As suggested by the reviewer, we replace the bidirectional LSTM encoder with the Conformer [Gulati et. al] model.  The Conformer-Transducer model uses an encoder with 10 Conformer blocks (512-dimensional feed-forward module, 31-kernel convolution block, and 8 64-dimensional attention heads). We train the model on the SWB 300H dataset for this experiment. All the other model architecture choices and hyper-parameters remain fixed.
>
> ***
>                       |       ATIS (WER)        |       HVB (WER)         |
>                       | Target      | Mixed     |   Target     | Mixed    |
> ***
>
>     Unadapted         |  11.5       |    11.9   |    39.7      |  26.0    |
>     NN-LM             |  11.3       |    11.9   |    38.1      |  25.6    |
>     Shallow Fusion    |  9.1        |    14.1   |    32.2      |  23.6    |
>     TOLSTOI           |  10.4       |    12.4   |    28.8      |  21.9    |
> ***
>
> We find that TOLSTOI is also effective with the Conformer encoder yielding improvements in WER on the mixed test sets and good improvements on the target-only test sets. We have added these results in the appendix of the paper.
>
> **Q2: Difference between learned imputation outputs and actual outputs from speech module.**
>
> **A2**: The imputation model is indeed discarded after the adaptation.
>
>
>
> The [tsne plot](https://freeimage.host/i/H9yTJbp) at the anonymised URL, compares the 256-dimensional representations obtained from (0) TOLSTOI and (1) actual audio from the SWB dataset corresponding to the same underlying text. We observe that there is a reasonable amount of overlap across both kinds of representations even with using a very lightweight imputation model.
>
> **Q3: Is “end-to-end” an appropriate explanation?**
>
> **A3**: We do not claim that TOLSTOI is end-to-end. The main point we want to highlight is that TOLSTOI offers in-situ adaptation of end-to-end RNN-T models, with accurate adaptation to the target domain, without introducing new layers or external LMs during deployment. We will make this point more explicit in the paper.

---

> > ### Comment · Reviewer_BcMJ · 2022-11-19
> > **Thank you for the authors' response**
> >
> > I appreciate your response and additional experiments. I believe these expanded results well demonstrate the effectiveness of the proposed method. Thank you.

---

### Official Review · Reviewer_hxLw · 2022-10-29

**Confidence:** 4
**Correctness:** 3
**Technical Novelty And Significance:** 3
**Empirical Novelty And Significance:** 3
**Recommendation:** 8

**Clarity, Quality, Novelty And Reproducibility:**

Clarity: clear enough
Novelty: good.
Reproducibility: easy if the author can make the code available.

**Strength And Weaknesses:**

Strength
1. Novelty; The idea proposed in this paper indeed meet the requirements mentioned in this paper for text-only domain adaptation, which are high accuracy, no retraining, no impact on inference speed and no deterioration on source domain. As far as I know, previous studies fail to meet all four requirements.
2. Intensive experiments and ablation studies show the effectiveness of the design choices.

Weakness
1. The dataset used in this paper is not large enough. There exists gigaspeech with 10k hours training data. With large dataset, it could relieve the overfitting on source domain data and provide better generalization ability, thus making the results more convincing.
2. Transducer based models are specific to ASR domain, thus it's hard to apply the method in this paper to other fields, such as speech translation/machine translation domain, making the contribution of this paper less accessible. If similar idea could be applied to LAS based ASR, it could be better. However, the current model design can only be used to transducer based ASR.

**Summary Of The Paper:**

This paper proposes a new method to do domain adaptation without textual data from target domain for Transducer based ASR. Text-only adaptation for ASR enables a compact model for target domain without relying on external LMs. The idea proposed in this paper is also novel and interesting. Experiments on a small and medium size ASR data shows the improvement over multiple related baselines while maintains the performance on source domain.

**Summary Of The Review:**

Given the strength and weakness, the reviewer likes the novelty and still intend to have this paper accepted.

---

> ### Author Response · Authors · 2022-11-19
> **Author's response to reviewer hxLw**
>
> We thank the reviewer for their insightful comments.
>
> **Q1: Performance of TOLSTOI when using large datasets with more than 10K hours of training data.**
>
> **A1**: We conduct experiments using an RNN-T model trained on SWB 2000H and additional proprietary data totaling close to 56K hours of labeled speech.
>
> ***
>                       |       ATIS (WER)       |       HVB (WER)         |
>                       | Target      | Mixed    |   Target    | Mixed     |
> ***
>
>     Unadapted         |  5.2        |    6.8   |   12.2      |  10.3     |
>     NN-LM             |  5.1        |    6.8   |   12.0      |  10.3     |
>     Shallow Fusion    |  2.8        |    7.1   |    8.6      |  12.2     |
>     TOLSTOI           |  2.8        |    5.9   |    7.9      |   8.7     |
> ***
>
> Even with an order of magnitude more training data, TOLSTOI performs the best compared to NN-LM and shallow fusion on both ATIS and HVB. Consistent with our results reported in the paper, TOLSTOI is also the least subject to catastrophic forgetting yielding the smallest mixed WERs. We have added these results in the appendix of the paper.
>
> **Q2: RNN-T models are specific to ASR. Can our ideas be applied to a LAS-based framework?**
>
> **A2**: Apart from ASR, RNN-T models have also been used for other tasks such as spoken translation [1], emotion recognition [2], language identification [2], etc. That said, the main ideas in TOLSTOI are not restricted to the RNN-T framework and can be incorporated within an encoder-decoder with attention (LAS-style) framework. The encoder and imputation model could stay the same as in our current setup. Using an appropriate training curriculum, the attention-based decoder could be retrained to use encoder states generated by speech from our pretraining data along with our imputed encoder states generated from text-only data. This is an interesting extension that we leave for future work.
>
>
> > References:
>
> [1] Liu, Dan, et al. "Cross attention augmented transducer networks for simultaneous translation." Proceedings of the 2021 Conference on Empirical Methods in Natural Language Processing. 2021.
>
> [2] Kons, Zvi, et al. "Extending RNN-T-based speech recognition systems with emotion and language classification." arXiv preprint arXiv:2207.13965 (2022).

---

### Decision · Program_Chairs · 2023-01-20

**Decision:**

Accept: poster

**Justification For Why Not Higher Score:**

The scope of the current paper is limited to transducer based ASR.

**Justification For Why Not Lower Score:**

The proposed method is novel and has been thoroughly justified.

**Metareview: Summary, Strengths And Weaknesses:**

Summary: the paper presents a novel text-only adaptation method for RNN-T based ASR system. It is a lightweight adaptation technique. Experimental results show that the proposed method achieves better quality to address domain differences.

Strengths: The idea is novel, detailed experiments and ablation studies are conducted to justify various design choices.

Weaknesses: The current algorithm design and experimental justification are only for the transducer based ASR.

**Note From Pc:**

if the above contains the word "oral" or "spotlight" please see: "oral" presentation means -> notable-top-5% and "spotlight" means -> notable-top-25%. As stated in our emails, we are disassociating presentation type from AC recommendations